# Imaging Aspects of Hepatic Alveolar Echinococcosis: Retrospective Findings of a Surgical Center in Turkey

**DOI:** 10.3390/pathogens11020276

**Published:** 2022-02-20

**Authors:** Mecit Kantarci, Sonay Aydin, Suat Eren, Hayri Ogul, Okan Akhan

**Affiliations:** 1Department of Radiology, Ataturk University, Yakutiye 25030, Turkey; drmecit@atauni.edu.tr (M.K.); drsuateren@atauni.edu.tr (S.E.); 2Department of Radiology, Erzincan Binali Yidirim University, Erzincan 24002, Turkey; 3Department of Radiology, Duzce University, Düzce 81620, Turkey; hayriogul@duzce.edu.tr; 4Department of Radiology, Hacettepe University, Ankara 06800, Turkey; akhano@tr.net

**Keywords:** alveolar echinococcosis, imaging, CT, MRI, US

## Abstract

Radiologists should be aware of the findings of alveolar echinococcosis (AE) due to the diagnostic and management value of imaging. We are attempting to define the most common diagnostic imaging findings of liver AE, along with the prevalence and distribution of those findings. The patients’ US, CT, and MRI images were reviewed retrospectively. CT images were acquired with and without the administration of contrast medium. The MRI protocol includes T2-weighted images (WI), diffusion (WI), apparent diffusion coefficient (ADC) maps, and pre- and post-contrast T1WIs. The current study included 61 patients. The mean age of the population was 58.2 ± 9.6 years According to Kratzer’s categorization (US), 139 lesions (73.1%) were categorized as hailstorm. According to Graeter’s classification (CT), 139 (73.1%) lesions were type 1-diffuse infiltrating. The most frequent types were Kodama type 2 and 3 lesions (MRI) (42.6% and 48.7%, accordingly). P2N0M0 was the most frequent subtype. The current study defines the major, characteristic imaging findings of liver AE using US, CT, and MRI. Since US, CT, and MRI have all been utilized to diagnose AE, we believe that a multi-modality classification system is needed. The study’s findings may aid radiologists in accurately and timely diagnosing liver AE.

## 1. Introduction

Cestodes belonging to the *Echinococcus* genus can cause Echinococcosis, a universal health issue. According to a recent paper, a consensus on nomenclature was established, and agreement on three names was reached: cystic echinococcosis (CE), alveolar echinococcosis (AE), *E. granulosus* sensu lato (SL), and neotropical echinococcosis (NE); all other names were rejected [1]. Echinococcus granulosus (*E. granulosus*) and Echinococcus multilocularis are both important for medical and public health because they cause cystic echinococcosis (CE) and alveolar echinococcosis, respectively (AE). CE and AE are both dangerous and severe disorders, with a high fatality rate and a dismal prognosis if not managed properly, particularly in the case of AE [2,3]. 

AE is a global disease that is most frequent in the northern hemisphere, particularly in Central and Eastern Europe, Turkey, and Russia, as well as northern Asia, which encompasses Japan, Alaska, North America, and China [4]. Worldwide, around 18,000 new cases of AE are reported each year. The larval mass in humans’ *E. multilocularis* and *E. granulosus* adult stages live in the intestines of carnivores (definitive hosts), primarily foxes and other wild canids and dogs for *E. multilocularis*, and primarily dogs for *E. granulosus*. The feces of carnivores are utilized to disperse eggs into the environment. When intermediate hosts such as wild and domestic herbivores and omnivores swallow eggs that contain an embryonic stage (oncosphere), the oncospheres penetrate the intestinal mucosa and invade the portal venous or lymphatic systems. The oncospheres then mature into a larval stage called metacestode within the target organ’s capillary bed (most commonly the liver and lung), where they gradually grow into a tumor-like parasitic tissue mass (*E. multilocularis*) or a cyst-like structure (*E. granulosus*). In humans, the larval mass of AE resembles a cancer in form and behavior because it proliferates indefinitely via exogenous budding and invades the surrounding tissue. The larva’s exogenous proliferation capacity enables it to generate metastases via the bloodstream in distant tissues such as bone, brain, and kidney. There is some evidence that AE may be transmitted by lymphatic drainage.

Hepatic AE is a rare but severe zoonosis.. The World Health Organization’s Informal Working Group on Echinococcosis devised a “PNM” categorization system to serve as a global standard for evaluating diagnostic performance and therapy outcomes. The PNM categorization system denotes the presence of a parasite mass in the hepatic artery (P), the involvement of surrounding organs (N), and the involvement of distant sites (M). The goal of the PNM classification system is to improve the quality of service and to allow for consistent outcome evaluation across healthcare institutions [5]. An important role for radiologists is to ensure timely referral to specialists and imaging follow-up; but because AE imaging features are so variable, initial misdiagnosis is common, especially in non-endemic areas. Because of atypical magnetic resonance imaging (MRI) presentation of hepatocellular carcinoma (HCC) and intra-hepatic cholangiocarcinoma (ICC), imaging findings of AE resemble some types of these tumors [6]. Imaging modalities such as US, computed tomography (CT), and MRI work well together to aid in the diagnosis of AE lesions, their morphology, and treatment options. For regular follow-up imaging in AE, ultrasonography serves as the first step in the screening procedure. Ultrasonography is also the primary diagnostic method, with specific serology confirming the diagnosis in 95% of cases. Typical AE calcifications can easily be seen on non-contrast-enhanced CT images; however, MRI defines the multi-vesicular AE lesions, necrosis presence, and intra- and extrahepatic invasion more accurately.

US surveys revealed unusually high CE and AE prevalences among asymptomatic individuals living in endemic areas, particularly among transhumant or nomadic pastoralists. CE and AE screening is justified since early detection results in a better prognosis following therapy. US implementation enabled a greater knowledge of the natural history of CE and AE, as well as the formation of a WHO-standardized taxonomy of CE cyst types [7,8]. Based on the general US appearance, a classification was previously suggested by Kratzer et al. According to this classification, lesions were classified as hailstorm (indistinct, irregular boundaries, non-homogeneous pattern, and hyperechoic formations, with or without dorsal acoustic shadow), pseudo cystic, hemangioma-like, ossification, metastasis-like (generally hypoechoic, these lesions have a common trait with typical hepatic metastases such as colorectal cancer: they lack the halo effect. Rather than that, a core, hyperechoic, non-homogeneous scar exists.) [9]. In 70% of patients, a typical US appearance is observed: AE lesions are often big in size. The lesion has uneven boundaries and a heterogeneous content, with patches of hyperechogenic (fibrous tissue) and hypoechogenic (“active” parasitic tissue) tissue juxtaposed. Frequently, hyperechogenic fibrous tissue includes calcifications that are easily detected by sonography as hyperechogenic lesions with distinctive dorsal shadowing [8]. Additionally, US can shed light on biliary and vascular involvement: dilatations of the intrahepatic bile ducts, as well as parasite tissue infiltration of the inferior vena cava, hepatic, or portal vein walls, are plainly visible [10].

Additionally, a classification based on CT scans was suggested by Graeter et al. This classification system is primarily based on the lesions’ morphology and degree of calcification: type 1 diffuse infiltrating, type 2 primarily circumscribed, tumor-like, type 3a primarily cystoid-intermediate, type 3b primarily cystoid-widespread, type 4 small-cystoid-metastatic, type 5 mainly calcified [11]. 

Kodama et al. proposed a more widely accepted classification based on MRI findings, consisting of five categories: multiple small round cysts without a solid component are classified as type 1 (4%); multiple small round cysts with a solid component are classified as type 2 (40%); a solid component surrounds a large and/or irregular pseudocyst with multiple small round cysts is classified as type 3 (46%); a solid component without cysts is classified as type 4 (4%); a large cyst without a solid component is classified as type 5 (6%) [12]. 

Radiologists should be familiar with the findings of AE because of the importance of imaging in the diagnosis and management of this entity. Familiarity with common imaging findings enables early and accurate diagnosis, thereby increasing the likelihood and success of surgical therapy. During the current study, we are attempting to define the most common diagnostic imaging findings of liver AE, along with the prevalence and distribution of those findings.

## 2. Materials and Methods

The local ethics committee approved this study, and informed consent was waived due to the retrospective nature. The examinations covered were conducted between January 2016 and December 2021. Examinations were obtained from a tertiary health care hospital. The data were saved in the electronic archives of the health care facility. All of the included images were acquired at the time of the initial diagnosis. Due to the fact that initial imaging findings were used, none of the patients received medical therapy prior to the imaging examinations.

Patients with hepatic AE who had at least two of the following features were included: (1) histological evidence of EM; (2) EM-specific serum antibodies discovered in a high-sensitivity blood test; (3) nucleic acid from EM detected in a clinical specimen. All of the included cases were confirmed AE cases, we did not involve any possible cases. A total of 61 patients were enrolled in the study.

The patients’ sonographic (US), CT, and MRI images were reviewed retrospectively, and findings were recorded. Three radiologists with a combined experience of 9 (S.A), 24 (M.K), and 35 (O.A) years in abdominal imaging examined the images. Consensus was used to determine the presence of the results; if a disagreement arose, the opinion of a fourth radiologist was used. The images were evaluated on a workstation, (Syngo.via, Siemens Healthineers, Erlangen, Germany). 

All US tests were conducted supine, all segments of the liver were documented, and lesions were also examined using color Doppler ultrasonography (CDUS). US results were classified according to Kratzer system [13]. 

Intravenous injection of 50–60 mL iohexol (4.0 mL/s) into the antecubital vein was followed by a 40-mL saline bolus for CT studies. Following scout acquisition, imaging was carried out in the supine posture, scanning in the cranio-caudal direction with the following parameters: 100–120 Sn kVp, 60–80 mAs, 0.33 s rotation time The thickness of the slices was 1.5 mm. Reconstruction of images was carried out in the axial, coronal, and sagittal planes. CT images were acquired with and without the administration of contrast medium. CT findings were classified according to Graeter et al.’s method [14].

The MRI protocol includes T2-weighted images (WI), diffusion-weighted images (WI), apparent diffusion coefficient (ADC) maps, and pre- and post-contrast T1WIs. For MRI findings, lesions were classified according to the Kodama classification system [15]. We determined ADCs of the solid components of AE lesions using a free-hand region of interest (ROI). One of the authors determined the ADC values for three different point of solid components and recorded the average of the three measurements as final data. 

We acquired imaging data from US, CT, and MRI tests, as well as age and gender information. PNM classification of the lesions was also noted. 

### Statistical Analysis

The Statistical Package for Social Sciences (SPSS) for Windows v.20 software was used to analyze the data (IBM SPSS Inc., Chicago, IL, USA). The Kolmogorov–Smirnov test was used to determine whether the data had a normal distribution. Numerical variables with normal distributions were represented by mean standard deviation values, while variables without normal distributions were represented by median (minimum–maximum) values. Percentages were used to represent categorical variables.

A two-tailed value of *p* < 0.05 was considered statistically significant.

## 3. Results

### 3.1. Patients

The current study included 61 patients, 35 (57.3%) of whom were female and 26 (42.6%) of whom were male. Mean age of the population was 58.2 ± 9.6 years, median age was 60 years (min.–max.; 43–77 years). All of the patients had US, CT, and MRI examinations. 

We were able to find treatment information for 58 patients; complete surgical excision and anthelmintic therapy were performed in 40 patients (68.9%). Fourteen patients (24.1%) had partial resection and anthelmintic therapy. Because the parasitic mass was unresectable, four (6.8%) patients underwent liver transplantation.

Table 1 illustrates the PNM stages of the patients and the details of the classification system.

### 3.2. US Results

According to US images, the mean number of AE liver lesions per patient was 3.1 ± 2.3 (range, 1–11; total lesion number, 190). The mean lesion size was 3.1 ± 2.8 cm (range, 1.1–15.2 cm). The lesion distribution was as follows: segment I (n = 29, 15.2%), segment II (n = 28, 14.7%), segment III (n = 8, 4.2%), segment IVa (n = 14, 7.3%), segment IVb (n = 15, 7.8%), segment V (n = 19, 10%), segment VI (n = 28, 14.7%), segment VII (n = 26, 13.6%), and segment VIII (n = 23, 12.1%). In total, 154 (81%) lesions were located in the right lobe.

A mass lesion with a mixed heterogeneous echogenic pattern and irregular contours, including cystic necrotic areas and multiple distributed calcific foci, was the most common sonographic appearance (73.1%) (Figure 1). Table 2 shows the distribution of sonographic appearances in great detail. Color Doppler ultrasonography (CDUS) revealed no vascularization in any of the lesions.

According to Kratzer’s categorization, 139 lesions (73.1%) were categorized as hailstorm, 4 (2.8%) as pseudo cystic, 41 (29.4%) as hemangioma-like, and 6 (4.3%) as ossification. 

### 3.3. CT Results

According to CT images, the mean number of AE liver lesions per patient was 3.8 ± 3.7 (range, 1–15; total lesion number, 232). The mean lesion size was 3.5 ± 2.2 cm (range, 1.1–18.8 cm). The lesion distribution was as follows: segment I (n = 33, 14.2%), segment II (n = 31, 13.3%), segment III (n = 11, 4.7%), segment IVa (n = 19, 8.1%), segment IVb (n = 21, 9%), segment V (n = 28, 12%), segment VI (n = 34, 14.6%), segment VII (n = 32, 13.7%), and segment VIII (n = 23, 9.9%). In total, 188 (81%) lesions were located in the right lobe.

The most frequently encountered CT pattern was a heterogeneous mass with calcifications and hypoattenuating areas (72.3%). Most of the lesions had irregular contours (85.4%), 85.4% lacked contrast enhancement, 77.4% of the lesions had calcifications, and atrophy and capsular retraction was present in 10.7% of the lesions (Figure 2 and Figure 3). Table 3 details the distribution of CT appearances.

According to Graeter et al.’s classification, 139 (73.1%) lesions were type 1-diffuse infiltrating, 41 (29.4%) lesions were type IV-small/cystoid-metastatic, 6 (4.3%) lesions were type V-mainly calcified, and 4 (2.8%) lesions were type 3b-primarily cystoid—widespread. 

### 3.4. MRI Results

According to MRI, the mean number of AE liver lesions per patient was 3.8 ± 3.7 (range, 1–15; total lesion number, 232). The mean lesion size was 3.5 ± 2.2 cm (range, 1.1–18.8 cm). The lesion distribution was as follows: segment I (n = 33, 14.2%), segment II (n = 31, 13.3%), segment III (n = 11, 4.7%), segment IVa (n = 19, 8.1%), segment IVb (n = 21, 9%), segment V (n = 28, 12%), segment VI (n = 34, 14.6%), segment VII (n = 32, 13.7%), and segment VIII (n = 23, 9.9%). In total, 188 (81%) lesions were located in the right lobe.

The most frequently encountered MRI pattern was a heterogeneous mass with irregular contours with central necrosis (78.7%). Most of the lesions (85.4%) lacked contrast enhancement (Figure 4). The most frequent types were Kodama type 2 and 3 lesions (42.6% and 48.7%, accordingly). Table 4 details the MRI findings.

The mean ADC value of hepatic AE lesions’ solid components was 1.51 ± 0.32 × 10^−3^ mm^2^/s (range: 0.97–1.78 × 10^−3^ mm^2^/s) (Figure 3). Table 5 summarizes the mean ADC values for the various types of AE lesions.

## 4. Discussion

In endemic areas, AE is a serious health problem and a diagnostic challenge for the radiologists as it can mimic hepatic malignancies and cystic liver diseases. We defined the major diagnostic findings of liver AE using US, CT, and MRI in the current study.

It was previously stated that liver AE is typically seen between the ages of 5–7 decades [13,16], which corresponded to the mean age of our population, which was in the fifth decade. 

According to our findings, the mean size of lesions was approximately 3 cm, and the majority of lesions were located in the right lobe. Similarly, previous research has indicated that lesions can grow to a diameter of up to 3 cm [13,14]. The distribution of lesions between the right and left lobes was also consistent with previous research [14,15,16]. Our findings indicated that the mean number of lesions per patient was approximately three; a similar finding was previously reported [15]. We have shown that US can detect fewer lesions than CT and MRI examinations. Confirming our findings, the literature indicates that US can be used as an initial investigative modality for alveolar echinococcosis detection; CT and MRI are more useful imaging methods [13,17].

Previously, typical hepatic AE findings in ultrasonographyincluded a large hepatic mass with juxtaposed areas of internal hyper- and hypoechogenicity, irregular margins, and scattered foci of calcification, as well as a pseudocyst with a large area of central necrosis surrounded by an irregular ringlike region of hyperechogenicity that resembled fibrous tissue. Additionally, no vascularization was anticipated during the CDUS examination [13]. Our findings were consistent with the literature; the majority of our patients exhibited the previously defined characteristic sonographic appearance, and CDUS revealed no vascularization. According to Kratzer et al. [9], the most common subtype was hailstorm pattern, while ossification and hemangioma-like subtypes were less common. Our examples were distributed in accordance with Kratzer’s classification. 

CT is emphasized as the primary imaging modality for determining the anatomic location and spread of lesions, as well as for characterizing lesions and detecting typical calcifications. Infiltrating tumorlike hepatic mass with irregular boundaries and heterogeneous contents with different degrees of attenuation, including scattered hyperattenuating calcifications and hypoattenuating regions related to necrosis and parasite tissue, were identified as typical AE findings [13,17]. Our most frequent CT appearance, consistent with the defined typical pattern, was a heterogeneous mass with calcifications and hypoattenuating areas. The majority of lesions had irregular margins, and calcifications were frequently observed. As for the enhancement characteristics, while contrast-enhanced CT does not reveal significant intralesional enhancement, delayed phase imaging does reveal mild enhancement in the peripheral fibro-inflammatory tissue [13]. Similarly, we detected no contrast enhancement within the lesions included; only perilesional enhancement was detected in a few lesions. We have encountered atrophy and capsular retraction in 10.7% of the lesions. Atrophy and capsular retraction have been observed in AE lesions as a result of vascular and biliary involvement [18]. When assessed according to the Graeter classification system [11], distribution of the cases was similar with the original paper: the most frequent subtypes were type 1 and type 4. 

Magnetic resonance imaging is the best modality for characterizing the components of parasite lesions [10]. On T1-weighted images, alveolar echinococcosis is distinguished by a heterogeneous infiltrative mass with irregular edges and a necrotic center, as well as heterogeneous signal intensity (areas of low and high signal intensity) on T2-weighted images [13,17]. In line with the literature, most of the included lesions (78.7%) were seen as a heterogeneous mass with irregular contours and central necrosis. Based on MRI findings, AE lesions are categorized under five subtypes [12], with type 2 and 3 being the most common [13]. Consistently with the literature, most of the included lesions were categorized under type 2 and 3 (42.6% and 48.7%, accordingly).

ADC values have recently been shown to be useful in differentiating AE lesions from simple cysts and malignant lesions. ADC values of AE lesions are higher than malignant lesions and lower than other hepatic cysts [13,15,19]. Mean ADC values of hepatic AE lesions was defined as 1.34 ± 0.41 × 10^−3^ mm2/s [19] and 1.73 ± 0.50 × 10^−3^ mm2/s [15]. Our values are comparable to those reported previously; we defined a mean value of 1.51 ± 0.32 × 10^−3^ mm^2^/s. 

Even though some classification systems were proposed for CT and US [9,11], they have not been widely used, yet. The MRI-based Kodama classification system [12] was more commonly accepted. Since US, CT, and MRI have all been utilized to diagnose AE, we believe that a multimodality classification system is needed. This assumes that the most powerful parts of all three systems are converged, such as CT for calcification detection and MRI for enhancement.

The current study has a few noteworthy limitations. Apart from the retrospective nature, the study’s population size is a constraint; additional studies with larger populations may alter the findings. While we have defined the imaging characteristics of hepatic AE lesions, we are unable to provide specific information about the imaging findings that distinguish AE from other liver lesions. Because the ADC values for the lesions were determined by a single researcher, we do not have data on interobserver variability. We give information about the imaging results of hepatic AE; however, because of the retrospective character of the study, we are unable to provide information about the clinical status of the included individuals.

## 5. Conclusions

Liver AE can present with a variety of imaging findings and can pose a diagnostic challenge for radiologists, particularly in areas where the disease is not prevalent. The current study defines the major, characteristic imaging findings of liver AE using US, CT, and MRI. The study’s findings may aid radiologists in accurately and timely diagnosing liver AE.

## Figures and Tables

**Figure 1 pathogens-11-00276-f001:**
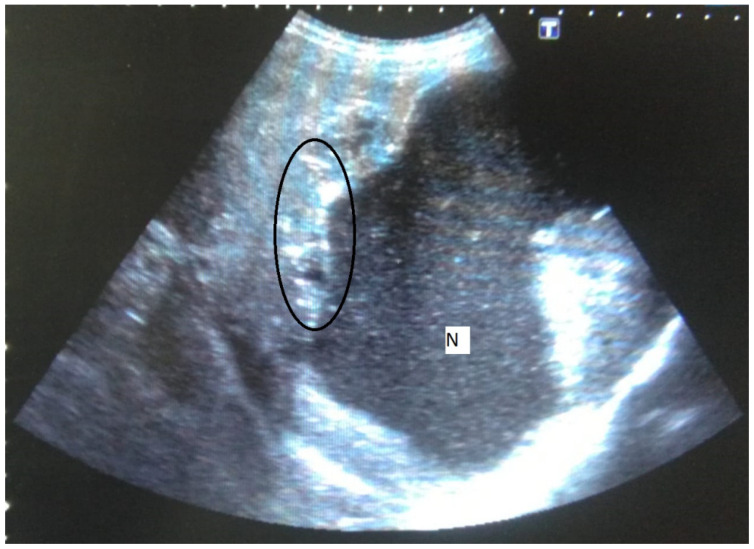
58-year-old female. Hepatic lesion with irregular margins, calcifications (circle), and large cystic-necrotic component (N) are seen.

**Figure 2 pathogens-11-00276-f002:**
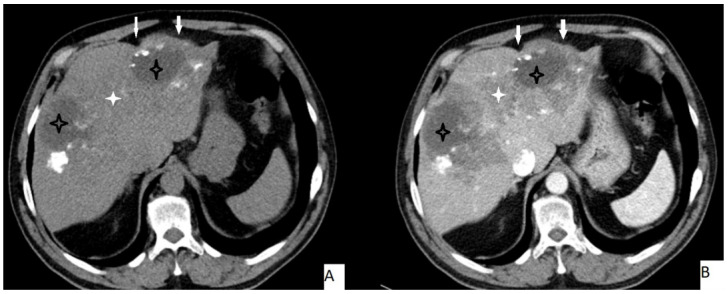
51-year-old male. Hepatic lesion with irregular contours and calcifications on precontrast axial CT (**A**). On portal venous images (**B**), no enhancement is present in the lesion. Central necrosis can be seen (**A**,**B**—black star). Perilesional enhancement (**B**—white star) and capsular (**A**,**B**—arrows) retraction are present.

**Figure 3 pathogens-11-00276-f003:**
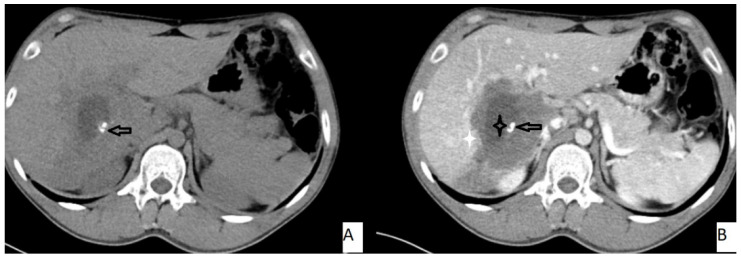
53-year-old female. Hepatic lesion with irregular borders and calcification (**A**,**B**—arrows) on precontrast axial CT (**A**). On portal venous images (**B**), no enhancement is present in the lesion. Central necrosis can be seen (**B**—black star). Perilesional enhancement (**B**—white star) is present.

**Figure 4 pathogens-11-00276-f004:**
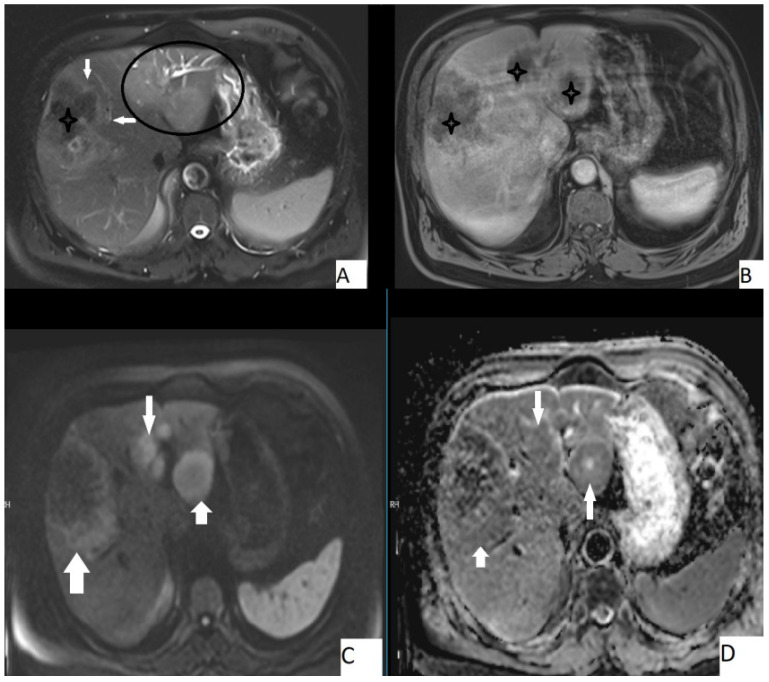
67-year-old male, axial T2WI (**A**), axial postcontrast T1WI (**B**), axial DWI (**C**), and axial ADC map (**D**). Hepatic lesion with irregular borders (**A**, arrows) and other hepatic homogeneous T2 hyperintense lesions (**A**, circle). Central necrosis was present (**A**,**B**, stars), no contrast enhancement is seen (**B**). Lesions were slightly hyperintense on DWI and slightly hypointense on ADC map. Mean ADC value of the large lesion was 1.58 × 10^−3^ mm^2^/s, while the smaller ones were 1.44 × 10^−3^ mm^2^/s and 1.42 × 10^−3^ mm^2^/s.

**Table 1 pathogens-11-00276-t001:** PNM status of the patients.

PNM System for Classification of Human Alveolar Echinococcosis		Number of Patients (%)
P	Hepatic localization of the primary lesion	
PX	Primary lesion cannot be assessed	-
P0	No detectable liver lesion	-
P1	Peripheral lesions without proximal vascular and/or biliary involvement	14 (22.9%)
P2	Central lesions with proximal vascular and/or biliary involvement of one lobe	32 (52.4%)
P3	Central lesions with hilar vascular and biliary involvement of both lobes and/or with involvement of two hepatic veins	11 (18%)
P4	Any lesion with extension along the portal vein, inferior vena cava, or hepatic arteries and the biliary tree	4 (6.5%)
N	Extra hepatic involvement of neighboring organs or tissues	
NX	Cannot be evaluated	-
N0	No regional involvement	55 (90.1%)
N1	Regional involvement of contiguous organs or tissues	6 (9.8%)
M	Absence or presence of distant metastases	
MX	Not completely evaluated	-
M0	No metastasis	57 (93.4%)
M1	Metastasis present	4 (6.5%)

**Table 2 pathogens-11-00276-t002:** Sonographic appearances of the lesions.

Sonographic Appearances	Number (%)
Mixed heterogeneous echogenic pattern and irregular contours, including cystic necrotic areas and multiple distributed calcific foci	139 (73.1)
Hailstorm pattern with multiple hyperechogenic solid lesions	41 (29.4)
Small calcified lesions	6 (4.3)
Pseudocyst with massive necrosis	4 (2.8)
Kratzer Types	
Hailstorm	139 (73.1)
Pseudo cystic	4 (2.8)
Hemangioma-like	41 (29.4)
Ossification	6 (4.3)

**Table 3 pathogens-11-00276-t003:** CT characteristics of the lesions.

CT Characteristics	Number (%)
Contour	
Irregular	182 (85.4)
Well-defined	31 (14.5)
Attenuation pattern	
Heterogenous with calcifications and hypoattenuating areas	154 (72.3%)
Homogeneous solid mass	43 (20.1%)
Mainly calcified mass	9 (4.2%)
Homogeneous cystic mass	7 (3.2%)
Contrast enhancement	
No enhancement	182 (85.4%)
Perilesional enhancement	31 (14.5%)
Calcification	
Present	165 (77.4%)
Absent	48 (22.5%)
Atrophy and capsular retraction	
Present	23 (10.7%)
Absent	190 (89.2%)
Graeter Types	
Type 1	139 (73.1%)
Type IV	41 (29.4%)
Type V	6 (4.3%)
Type 3b	4 (2.8%)

**Table 4 pathogens-11-00276-t004:** MRI characteristics of the lesions.

MRI Characteristics	Number (%)
Contour	
Irregular	182 (85.4)
Well-defined	31 (14.5)
Internal intensity	
Heterogenous	163 (76.5%)
Homogeneous	50 (23.3%)
Contrast enhancement	
No enhancement	182 (85.4%)
Perilesional enhancement	31 (14.5%)
Kodama types	
Type 1	3 (1.2%)
Type 2	99 (42.6%)
Type 3	113 (48.7%)
Type 4	13 (5.6%)
Type 5	4 (1.7%)
Atrophy and capsular retraction	
Present	23 (10.7%)
Absent	190 (89.2%)

**Table 5 pathogens-11-00276-t005:** Mean apparent diffusion coefficient values for alveolar echinococcosis lesions.

Kodama Types	Mean Apparent Diffusion Coefficient Value (×10^−3^ mm^2^/s) (Mean ± SD)
Type 1	1.92 ± 1.01
Type 2	1.78 ± 0.86
Type 3	1.57 ± 0.11
Type 4	1.15 ± 0.21
Type 5	1.9 ± 0.18

## Data Availability

Available upon request.

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
