# Peer review of "Imaging Aspects of Hepatic Alveolar Echinococcosis: Retrospective Findings of a Surgical Center in Turkey"

_pathogens, 2022, doi:10.3390/pathogens11020276_

Round 1

Reviewer 1 Report

The present revision has included most of the suggestions. There are still some points to be covered (lines provided):

Abstract: It is suggested to include the comparison of the three imaging classifications as well as PNM, as this is a highlight of the study! How they match and perform, etc.

Intro
22: first sentence is imprecise, it is not one disease, but several! This was addressed in the paper of Kern etal in Advances Parasitol. Modify to echinococcoses, and zoonotic diseases that are....
26: Add E.granulosus sensu lato (s.l.), as more and more species are being discovered, which is -by the way -content of the citation #2!
29: citation #2 is given to the diseases in animals, but not humans. Instead, Kern et al in the Advances (co-author is Okan Akhan!!) should be cited.
46-8: Brunetti et al must be cited at this moment. You also can refer to the PNM paper by Kern et al,2006  as both are the basis for the mentioned statement.
64: Barth et al described a immunohistological study, but nothing about imaging. Thus, I wonder why you put in this citation? might be more valid in the following parts regarding confirmed diagnosis of AE etc. e.g. in Materials section.
74: it is suggested to name Kratzer (and later Graeter) explicitly, as you refer to their work (which are the results of a >20y long experience). Note, you mention Kodama who published his work >15y ahead of Kratzer and Graeter. No wonder, why he received more attention as time went by! It seems that classifying MRI is much easier (and as the method is your favored one) as US or CT?? By the way, you mentioned all three authors in the MM section.
99: Wouldn't you think that you also validated the published classification of Kratzer, Ggraeter, Kodama?? and in your case all pts were treatment naive!
Materials and Methods
109: You were lucky, all pts underwent immediate surgery, thus histo or PCR in addition to serology are available. For "naive" readers and imagers of new cases it would be helpful to approach a case according to the suggestions in Brunetti et al, possible, probable, confirmed?? Differential diagnosis is still the major issue. Maybe your final statement in line 297 should have a reflection at this section already??
Results
Findings were categorized according to previous proposals of Kratzer, Graeter and Kodama. It would be nice to see the classifications of kratzer and Graeter in the tables, as it was displayed for Kodama in table 3?
150ff: This paper is on radiology, not treatment. findings are irrelevant for the topic of imaging tools. The given figures about treatment have no impact for the present study but are even misleading as nothing is said about the treatment strategy of the center, postop care, etc. It is suggested to delete this paragraph. See also line 296, your last sentence.

Discussion
237: "we have shown.." Check for minor English style in the entire manuscript.
240: check reference #5 is your own group. others are reviews. and literature is full of original publications from many countries.
248: it seems that "literature" concerns the own previous publications. #5,#9,and #17 are repeatedly mentioned but work of others is not mentioned appropriately.
269: sentence is unclear. Do findings represent the distribution provided by Graeter? Confirm the proposal? or does it mean that you validated the class. by Graeter?? Which would be important to stress in the Abstract. Readers and imagers would be guided to go for??
285: sentence is now inconsistant with the findings above. You assigned already a value on both US and CT classifications?? Alter the sentence.
288: Would you envisage such a multimodality classification?? Is it perhaps work in progress? Would be great to provide some ideas?? how to converge the present systems??

Author Response

Abstract: It is suggested to include the comparison of the three imaging classifications as well as PNM, as this is a highlight of the study! How they match and perform, etc.

Response: The abstract was revised according to the comments: “Radiologists should be familiar with the findings of Alveolar Echinococcosis (AE) because of the importance of imaging in the diagnosis and management. We are attempting to define the most common diagnostic imaging findings of liver AE, along with the prevalence and distribution of those findings. The patients' US, CT, and MRI images were reviewed retrospectively. CT images were acquired with and without the administration of contrast medium. The MRI protocol in-cludes T2-weighted images (WI), diffusion (WI), apparent diffusion coefficient (ADC) maps, and pre- and post-contrast T1WIs. The current study included 61 patients. Mean age of the population was 58.2±9.6 years According to Kratzer's categorization (US), 139 lesions (73.1 %) were catego-rized as hailstorm. According to Graeter’s classification (CT), 139 (73.1%) lesions were type 1-diffuse infiltrating. Most frequent type was Kodama type 2 and 3 lesions (MRI) (42.6% and 48.7%, accordingly). P2N0M0 was the most frequent subtype.. A mass lesion with a mixed het-erogeneous echogenic pattern and irregular contours, including cystic necrotic areas and multiple distributed calcific foci, was the most common sonographic appearance (73.1%). The most fre-quently encountered CT pattern was a heterogeneous mass with calcifications and hypoattenu-ating areas (72.3%). The most frequently encountered MRI pattern was a heterogeneous mass with irregular contours with central necrosis (78.7%). The current study defines the major, char-acteristic imaging findings of liver AE using US, CT, and MRI. Since US, CT and MRI have all been utilized to diagnose AE, we believe that a multi-modality classification system is needed. The study's findings may aid radiologists in accurately and timely diagnosing liver AE.”

Intro
22: first sentence is imprecise, it is not one disease, but several! This was addressed in the paper of Kern etal in Advances Parasitol. Modify to echinococcoses, and zoonotic diseases that are....

Response: The mentioned sentence was revised as: “Cestodes belonging to the Echinococcus genus can cause Echinococcosis, a universal health issue.”

26: Add E.granulosus sensu lato (s.l.), as more and more species are being discovered, which is -by the way -content of the citation #2!

Response: It was added.

29: citation #2 is given to the diseases in animals, but not humans. Instead, Kern et al in the Advances (co-author is Okan Akhan!!) should be cited.

Response: The mentioned reference was added as number 3: “Kern, P., Da Silva, A. M., Akhan, O., Müllhaupt, B., Vizcaychipi, K. A., Budke, C., & Vuitton, D. A. (2017). The echinococcoses: diagnosis, clinical management and burden of disease. In Advances in parasitology (Vol. 96, pp. 259-369). Academic Press.”

46-8: Brunetti et al must be cited at this moment. You also can refer to the PNM paper by Kern et al,2006  as both are the basis for the mentioned statement.

Response: The references were revised as recommended.

64: Barth et al described a immunohistological study, but nothing about imaging. Thus, I wonder why you put in this citation? might be more valid in the following parts regarding confirmed diagnosis of AE etc. e.g. in Materials section.

Response: The mentioned reference was removed.

74: it is suggested to name Kratzer (and later Graeter) explicitly, as you refer to their work (which are the results of a >20y long experience). Note, you mention Kodama who published his work >15y ahead of Kratzer and Graeter. No wonder, why he received more attention as time went by! It seems that classifying MRI is much easier (and as the method is your favored one) as US or CT?? By the way, you mentioned all three authors in the MM section.

Response: The relevant section was revised as recommended.

99: Wouldn't you think that you also validated the published classification of Kratzer, Ggraeter, Kodama?? and in your case all pts were treatment naive!

Response: The similarities of the study population with Kratzer, Graeter, Kodama were emphasized in the discussion section, as recommended.

Materials and Methods
109: You were lucky, all pts underwent immediate surgery, thus histo or PCR in addition to serology are available. For "naive" readers and imagers of new cases it would be helpful to approach a case according to the suggestions in Brunetti et al, possible, probable, confirmed?? Differential diagnosis is still the major issue. Maybe your final statement in line 297 should have a reflection at this section already??

Response: Thank you very much for the comment. As we work in an endemic area a lot of AE are being referred in our center, as the reviewer emphasized we tried to share our experiences to create awareness for the typical imaging findings of hepatic AE.

Results
Findings were categorized according to previous proposals of Kratzer, Graeter and Kodama. It would be nice to see the classifications of kratzer and Graeter in the tables, as it was displayed for Kodama in table 3?

Response: The table 1 and 2 were revised as recommended.

150ff: This paper is on radiology, not treatment. findings are irrelevant for the topic of imaging tools. The given figures about treatment have no impact for the present study but are even misleading as nothing is said about the treatment strategy of the center, postop care, etc. It is suggested to delete this paragraph. See also line 296, your last sentence.

Response: the relevant sections were omitted.

Discussion
237: "we have shown.." Check for minor English style in the entire manuscript.

Response: The manuscript was checked for the minor errors.

240: check reference #5 is your own group. others are reviews. and literature is full of original publications from many countries.

Response: the mentioned reference was omitted.

248: it seems that "literature" concerns the own previous publications. #5,#9,and #17 are repeatedly mentioned but work of others is not mentioned appropriately.

Response: Reference 5 was omitted, and the other related studies were discussed.

269: sentence is unclear. Do findings represent the distribution provided by Graeter? Confirm the proposal? or does it mean that you validated the class. by Graeter?? Which would be important to stress in the Abstract. Readers and imagers would be guided to go for??

Response: The findings were similar with Graeter , Sade, and Becce et al.’s results. The relevant paragraph was revised in a more clear manner, and also a reference representing Graeter’s study was added.

285: sentence is now inconsistant with the findings above. You assigned already a value on both US and CT classifications?? Alter the sentence.

Response: the paragraph was revised as recommended:” Previously, typical US findings of hepatic AE included a large hepatic mass with juxtaposed areas of internal hyper- and hypoechogenicity, irregular margins, and scattered foci of calcification, as well as a pseudocyst with a large area of central necrosis surrounded by an irregular ringlike region of hyperechogenicity resembling fibrous tissue. Additionally, no vascularization was anticipated during the CDUS examination [6,18]. Our findings were consistent with the literature; the majority of our patients exhibited the previously defined characteristic sonographic appearance, and CDUS revealed no vascularization. According to Kratzer et al. [14], the most common subtype was hailstorm pattern, while ossification and hemangioma-like subtypes were less common. Our examples were distributed in accordance with Kratzer's classification.”

288: Would you envisage such a multimodality classification?? Is it perhaps work in progress? Would be great to provide some ideas?? how to converge the present systems??

Response: “Even though some classification systems were proposed for CT and US [14,16], they have not been widely used, yet. MRI based Kodama classification system [17] was more commonly accepted.[17] was more commonly accepted. Since US, CT and MRI have all been utilized to diagnose AE, we believe that a multimodality classification system is needed. Assume that the most powerful parts of all three systems are converged, such as CT for calcification detection and MRI for enhancement.” The paragraph was added into the discussion.

Reviewer 2 Report

The manuscript shows a significant improvement towards the proposed research objective. Great job.

Author Response

Thank you for the encouraging comments.

Best regards

This manuscript is a resubmission of an earlier submission. The following is a list of the peer review reports and author responses from that submission.

Round 1

Reviewer 1 Report

This review refers to the Article with the title “Imaging Findings of Hepatica Alveolar Echinococcosis” submitted for publication in Pathogens. The main purpose of the manuscript is to identify and characterize the most common diagnostic imaging findings of liver Alveolar Echinococcosis (AE) based on a retrospective review of US, CT, and MRI exams. The claimed pertinence is for the target audience, radiologists, to be familiar with this pathology. Overall, the manuscript shows the expected scientific presentation, using an appropriated English language, correct style with a good readability level. The paper works as a refresher course on the main findings of the referred disease, colleting different findings in different image modalities. Medical community and in particular junior radiologists and radiographers may benefit from it.

The major weakness of the paper, in the reviewer opinion, is the description of the methods followed by the authors, in particular:

  1. Line 71 – (…) retrospective nature. What was the time range? From where the data was collect?
  2. Line 76 – (…) images were reviewed retrospectively and findings were recorded. Who did the reporting of the exams? How expert were they? Only one examiner or more? What type of equipment was used to perform the exam evaluation with the required clinical quality settings?
  3. What the type of exams, sequences (MRI), acquisition parameters (CT and US), post processing techniques did the included exams had? Where there significant differences?
  4. How was the data stored? Where?
  5. What type of variables were stored? Data from other clinical methods and from the patient story was collected? If so please state how may affect the imaging patterns presented.

Reviewer 2 Report

Authors describe imaging findings of 61 patients with alveolar echinococcosis. The analysis was performed unicentrically, and retrospectively while using available records of images as well as of reports.  All patients underwent surgery and received benzimidazoles for treatment while 4 of the patients had liver transplantation. But treatment is not at all in focus of this paper instead the detailed radiomorphological findings of surgical patients, possibly before any treatment. Thus, introduction and discussion may further concentrate on imaging findings. The value of this paper is fine description of liver lesions obtained with three different methods. The paper would gain actuality if authors would go for the classifications of US (Kratzer) and CT (Graeter) similarily to the MRI of Kodoma, all of those mentioned in the Introduction. Furthermore, it is compulsory to mention the WHO recommendations for diagnosis and treatment (Brunetti et al), as these consensus statements can be regarded as the present standard.

The patient description is insufficient. A new table should provide all (even more than present) demographic data. How and when were the patients recruited? Imaging data were used at the time of diagnosis? or during the chronic state of the diseases, months or years after first diagnosis, etc.? In addition, the PNM status could be assigned in order to compare the data to those of other investigators as well as to indicate the extend of the AE disease. All of this could be part of a new table.

16 references were cited, and the majority thereof refer strictly to imaging findings and are displayed as reviews or concluding observations (#3,5,9,10,12,15,16) or propositions for classification (#6,7,8) on the basis of the radiological findings (i.e. US, CT, MRT, PET/CT, etc.). This restriction may be appropriate, but limits the depth of discussion. For a paper exclusively dedicated to the radiomorphological findings it is suggested to cite further primary work regarding radiomorphology, and there are many!

The variability of liver lesions is displayed. Many previous reports in the past have had the same impetus. It would have been interesting to see whether the data would match with others', or would fit into classifications for US of Kratzer or CT of Graeter, as contrasted to the MRI classification of Kodoma? Admittedly, it may be difficult for US in retrospect, but it can be assumed that the first authors are those who did all US exams over time and can abstract from their experiences? 

It is suggested to narrow the title in "Imaging aspects......: Retrospective findings of a surgical center in Turkey". This would make clear that the data are based upon a single-centre experience.

Authors have used the data set of a surgical center where patients were refered for excision of lesions. Since many years it is a consensus among international specialists that palliative treatment of AE has to be disfavored. Liver transplantation is a last resort, and has strong limitations. Furthermore, in centres with high expertise, LTx needs rarely to be applied. If authors would have considered the huge amount of international knowledge about they would have modified the herein described opinions and speculations. Unless authors reshape the article entirely, all parts concerning therapy should be omitted since the corresponding data from the patient group are not presented. 

Comments to some of the statements (still incomplete, as to my impression the paper needs a deep revision, lines are indicated):
22: consider terminology!! E. granulosus s.l. There are many more! see Vuitton et al 2020, Romig et al and Deplazes et al all in Adv Parasitol.
25: Paper #1 describes animals not humans! use Kern, .. Okan Akhan(one of the present co-authors!!!).. in Advances in Parasitology. The cited paper #1 is misplaced, as it deals with dogs and foxes.
28: Arrangement of countries is strange. China is the hot spot, in North America you can count the number of human cases at one hand! But it is correct for the natural habitat, E. multilocularis is endemic in North America! Is the message directed to clinicians or to biologists? Possibly, refer to Deplazes in Adv.Parasitol, see above.
34: Buttenschoen et al,  Barth et al, Hillenbrand et al., Brehm et al worked on this particular aspect. You may find even more primary publications. Lymphatic drainage is not a mean of transmission! Pls correct the statement or approve by correct citations.
35: Statement is incorrect. Consider PNM classification, which needed to be referenced at this stage. Also statements of the next sentences need to be supported by citations, e.g. Peters et al J Hepatol 2021, Bresson-Hadni et al. 2021. Pls check for further fine tuning of your statements.
44: Consider e.g. the European and Asian surveys with US, all were based upon US findings and have described and explained the morphology. 
46: consider the French literature, Bartholomot was the first author to describe the fine structure of AE lesions! 
50: CE classification is not referenced, pls check and cite Brunetti et al, and the corresponding anonymous WHO paper.
72ff: I wonder how you came to the diagnosis when your first item is the histopathology? refer to the consensus recommendations Brunetti et al. and Kern et al. Add figures for patients with possible, probable, and confirmed diagnosis. Provide further info about the patient group, when diagnosed, how long been treated with albendazole prior to imaging, and many more  details. Provide a separate table of the patient group as suggested above. The preponderance of female patients might be due to the selection process for surgery, and is meaningless when considering a rather small group of 61 patients. Pls provide info how patient were recruited for surgery etc. and whether patients were kept on benzimidazoles only. (Check corresponding paper of Grüner et al. 2017). 
Treatment and follow-up is not an objective of this analysis. Those aspects might be included, in particular as authors spend a lot of effort to discuss about. For the present text therapeutical conclusions and speculations are premature and may be not balanced according to the international standards. It is suggested to present a further analysis in a separate paper.
170ff: It is suggested to entirely restructure the Discussion, see statements above.